# Trastuzumab and Doxorubicin Sequential Administration Increases Oxidative Stress and Phosphorylation of Connexin 43 on Ser368

**DOI:** 10.3390/ijms23126375

**Published:** 2022-06-07

**Authors:** Michela Pecoraro, Stefania Marzocco, Silvia Franceschelli, Ada Popolo

**Affiliations:** Department of Pharmacy, University of Salerno, 84084 Fisciano, Italy; mipecoraro@unisa.it (M.P.); smarzocco@unisa.it (S.M.); email@gmail.com (S.F.)

**Keywords:** Connexin 43, Doxorubicin, Trastuzumab, cardiotoxicity, calcium homeostasis

## Abstract

Human epidermal growth factor receptor-2 (HER2) is overexpressed in up to 30% of breast cancer cases, causing a more aggressive tumour growth and poor prognosis. Trastuzumab, the humanized antibody targeted to HER2, increased the life expectancy of patients, but severe cardiotoxicity emerged as a long-term adverse effect. Clinical evidence highlights that Trastuzumab-induced cardiotoxicity drastically increases in association with Doxorubicin; however, the exact mechanisms involved remain incompletely understood. In order to analyse the molecular mechanisms involved and the possible adaptative responses to Trastuzumab and Doxorubicin treatment, in this study, H9c2 cardiomyoblasts were used. Results showed that Trastuzumab and Doxorubicin sequential administration in cardiomyoblast increased cytosolic and mitochondrial ROS production, intracellular calcium dysregulation, mitochondrial membrane depolarization, and the consequent apoptosis, induced by both Trastuzumab and Doxorubicin alone. Furthermore, in these conditions, we observed increased levels of Connexin43 phosphorylated on Ser368 (pCx43). Since phosphorylation on Ser368 decreases gap junction intracellular communication, thus reducing the spread of death signals to adjacent cells, we hypothesized that the increase in pCx43 could be an adaptative response implemented by cells to defend neighbouring cells by Trastuzumab and Doxorubicin sequential administration. However, the other side of the coin is the resulting conduction abnormalities.

## 1. Introduction

Breast cancer is the most common type of cancer in women in the United States [1] and worldwide [2].

In up to 30% of breast cancer, an overexpression of the human epidermal growth factor receptor-2 (HER2) is found, with consequent aggressive tumour growth and poor prognosis. Trastuzumab (Tzb) was approved by the FDA as the first HER2-targeted therapeutics for treating patients with HER2-positive breast cancer, and its clinical use has led to a significant decline in breast cancer mortality [3]. However, the increase in survivors has allowed one to observe long-term side effects of chemotherapeutic drugs. Among these side effects, cardiotoxicity is the one that occurs most frequently. Cardiovascular disease is now the second leading cause of long-term morbidity and mortality among cancer survivors and the first cause of death among female survivors from breast cancer [4]. 

The main signalling pathways mediated by HER2 involve rat sarcoma (RAS)/rapidly accelerated fibrosarcoma (RAF)/mitogen-activated protein kinase (MAPK) pathways and phosphoinositide-3-kinase (PIK3)/protein kinase B (Akt)/mechanistic target of rapamycin (mTOR) kinase pathways, which regulate growth, proliferation, differentiation, motility, and adhesion of normal cells [5]. HER2 is thought to participate in an important pathway for growth, repair, and survival of adult cardiomyocytes [6,7,8]; thus, inhibition of HER2 in cardiomyocytes increases oxidative and nitrosative stress and reduces the prosurvival signalling pathways [9]. 

Clinically, Tzb-induced cardiotoxicity is characterised by symptomatic chronic heart failure (CHF) or asymptomatic left ventricular ejection fraction (LVEF) decline that can cause severe cardiac insufficiency and even death. It has been shown that about 4% of women with breast cancer develop heart failure during Tzb treatment, and this figure can reach 27% when Tzb is combined with Doxorubicin (Doxo). Tzb as a single therapy has a scarce antitumor activity and needs to be associated with a high cytotoxic drug such as anthracycline [10]. To minimize the risk of cardiotoxicity, Doxo and Tzb are not administered in combination therapy, but sequentially [11]. However, even this therapeutic strategy does not eliminate the cardiotoxic potential of the two drugs. It has been shown that Doxo administration induces HER2 overexpression in cardiomyocytes, possibly as an adaptive response to allow survival under stress conditions [12]. As a consequence, the pool of cardiomyocytes that depend on functional HER2 signalling for recovery rises [13], and thus, it has been hypothesized that the subsequent administration of Tzb, by inhibiting HER2, reduces the protection from mitochondrial damage in cardiomyocytes, leading to apoptosis and thus to cardiac dysfunction [14]. 

Despite that many clinical data are available regarding Tzb plus Doxo induced cardiotoxicity, the precise mechanisms underlying this effect remain poorly understood [15]. 

At the molecular level, Doxo-induced cardiotoxicity seems to be related to increased oxidative and nitrosative stress, calcium homeostasis dysregulation and mitochondrial dysfunction [16], while a recent study showed that Tzb exerts its cardiotoxic potential by impacting mitochondrial function [17]. Furthermore, it has been shown that in association with Doxo, Tzb exacerbates antioxidant enzyme pool damage and mitochondrial dysfunction [18]. 

Connexin 43 (Cx43) is a member of the connexin family, proteins that form gap junctions that mediate electrical and chemical signalling throughout the cardiac system and thus enable a synchronized contraction. Moreover, it can form hemichannels at the intercalated discs, at membrane lateral borders and at mitochondrial membranes. Hemichannels regulate extracellular communications, especially allowing for the dissemination of disease, since gap junctions and connexins have been reported to conduct both cell survival and cell death signals [19]. Many studies showed that Doxo administration induces alterations in Cx43 localization and activity in cardiac cells [20,21], and similar results have been recently reported for other chemotherapeutic drugs such as Ponatinib [22] and Tzb [17]. 

Currently, the most commonly used methods to detect chemotherapeutic drugs-induced cardiotoxicity are the evaluation of functional parameters that, unfortunately, are often detected only after considerable cell loss has taken place. Studies on molecular mechanisms responsible for functional impairment observed in the myocardium could help to identify therapeutic strategies to reduce this serious side effect. To this aim, in this study, we evaluated the effects of Doxo and Tzb sequential administration on H9c2 cells on oxidative stress and apoptosis and the involvement of Cx43.

## 2. Results

### 2.1. Effect of Doxorubicin, Trastuzumab and Their Association on Cytosolic and Mitochondrial ROS Production

Cytosolic ROS production was evaluated by means of the fluorescent probe DCHF-DA. Our data showed that in control cells, ROS levels were 32.44 ± 5.07. These levels were significantly higher both in Doxo-(51.93 ± 2.18; *p* < 0.05) and in Tzb-treated cells (51.26 ± 4.36; *p* < 0.05) compared to control cells. Instead, in cells treated first with Doxo and subsequently with Tzb, ROS production was significantly higher (62.62 ± 1.53; *p* < 0.005) than control cells (Figure 1A). 

Data obtained by means of MitoSOX red showed that in control cells, mitochondrial ROS levels were 19.07 ± 0.98. A significant increase in mitochondrial ROS production was observed in Doxo-treated cells (33.95 ± 2.72; *p* < 0.005 versus control cells). In Tzb-treated cells, mitochondrial ROS production was significantly higher (29.0 ± 1.79; *p* < 0.05) than control cells, but the increase is less evident than in cells treated with Doxo alone. In cells previously treated with Doxo and then with Tzb, mitochondrial ROS production was significantly higher (34.6 ± 2.11; *p* < 0.005) than control cells, likewise observed in cells treated with Doxo alone (Figure 1C). 

### 2.2. Effect of Doxorubicin, Trastuzumab and Their Association on Mitochondrial Membrane Depolarization

Mitochondrial damages occur as a consequence of increased ROS production, resulting in mitochondrial membrane integrity disruption and loss of its negative charge. In order to analyse the state of depolarization of the mitochondrial membrane in our experimental conditions, we used the fluorescent dye TMRE. Since TMRE is a cationic dye, it crosses only the membrane of healthy mitochondria, is negatively charged, and emits fluorescence, while in damaged mitochondria, as a consequence of depolarization, TMRE is not trapped, and the fluorescence intensity decreases. 

In control cells, the percentage of TMRE-positive cells was 67.89 ± 6.24. Our results showed a significant decrease in the percentage of TMRE-positive cells in Doxo-treated cells (31.8 ± 1.19; *p* < 0.001 versus control cells). In addition, in Tzb-treated cells, the percentage of TMRE-positive cells was significantly lower (38.21 ± 5.48; *p* < 0.005) than control cells, but mitochondrial membrane depolarization was more evident in Doxo and Tzb sequentially treated cells, since in these experimental conditions, the percentage of TMRE-positive cells appeared to be significantly lower (25.45 ± 1.99; *p* < 0.0001) than control cells (Figure 2).

### 2.3. Effect of Doxorubicin, Trastuzumab and Their Association on Calcium Homeostasis

Intracellular calcium concentrations were evaluated by means of FURA 2-AM in calcium-free incubation medium (containing 0.5 mM EGTA). Mitochondrial calcium content was evaluated by means of the mitochondrial calcium depletory, carbonyl cyanide p-trifluoromethoxyphenylhydrazone FCCP (50 nM). As reported in Figure 3A, the percentage of delta increase in mitochondrial calcium levels was 5.8 ± 0.92 in control cells. Both in Doxo- and in Tzb-treated cells, a reduction, despite not being significant, in the percentage of delta increase in mitochondrial calcium levels was observed compared to control cells. In Doxo and Tzb sequentially treated cells, the percentage of delta increase in mitochondrial calcium levels was significantly lower (3.9 ± 0.03; *p* < 0.005) than control cells. Data obtained by means of Ionomycin showed that in control cells, the delta increase in intracellular calcium levels was 5.9 ± 0.5. Doxo administration did not affect the percentage of delta increase in intracellular calcium levels (5.8 ± 0.2), while a significant reduction was observed in Tzb-treated cells (2.4 ± 0.3; *p* < 0.005 versus control cells and *p* < 0.05 versus Doxo-treated cells). In Doxo and Tzb sequentially treated cells, a reduction in the percentage of delta increase in intracellular calcium levels (3.9 ± 0.03), despite not being significant, was observed compared to control cells, indicating higher basal levels of calcium in these experimental conditions (Figure 3B). 

### 2.4. Effect of Doxorubicin, Trastuzumab and Their Association on Apoptosis 

Increased levels of ROS within the mitochondria can damage the mitochondrial DNA, thus leading to apoptosis and cytochrome c release from the mitochondria. We evaluated apoptosis by cytofluorimetric analysis of PI-stained hypodiploid nuclei. Our results indicated that in control cells, the percentage of hypodiploid nuclei was 8.66 ± 0.43. Doxo administration significantly induced the apoptotic response, as shown by the percentage of the hypodiploid nuclei (32.79 ± 2.64; *p* < 0.001 versus control cells). In addition, in Tzb-treated cells, a significant increase in the apoptotic response was observed (28.61 ± 2.61; *p* < 0.005 versus control cells), but this effect was significantly higher in Doxo and Tzb sequentially treated cells (31.76 ± 3.76; *p* < 0.001 versus control cells) (Figure 4A).

As reported in Figure 4C, our data showed that Tzb alone did not significantly affect cytochrome c release. Cytosolic cytochrome c levels were 33.24 ± 2.49 in control cells and 44.21 ± 4.07 in Tzb-treated cells. On the contrary, a significant increase in cytochrome c release was observed in cells treated with Doxo alone (68.01 ± 3.31; *p* < 0.005 versus control cells and versus Tzb-treated cells) and in cells sequentially treated with Doxo and Tzb, (66.24 ± 3.75; *p* < 0.005 versus control cells). 

### 2.5. Effect of Doxorubicin, Trastuzumab and Their Association on Cx43 and pCx43 Expression

Cx43 and Cx43 phosphorylated on Ser368 (pCx43) levels on the plasma membrane were evaluated by means of cytofluorimetry in non-permeabilized conditions. Our data showed that none of the treatments used affect the levels of Cx43 on the plasma membrane. The membrane levels of Cx43 were 55.35 ± 4.56 in control cells, 42.92 ± 6.17 in Doxo-treated cells, 60.07 ± 0.58 in Tzb-treated cells and 56.51 ± 1.98 in Doxo and Tzb sequentially treated cells (Figure 5A).

As reported in Figure 5C, pCx43 levels were 35.68 ± 1.84 in control cells. Doxo administration significantly affected pCx43 levels (64.53 ± 3.53; *p* < 0.005 versus control cells and *p* < 0.05 versus Tzb-treated cells), while no significant differences were observed in Tzb-treated cells (44.92 ± 4.62) compared to control cells. Instead, in Doxo and Tzb sequentially treated cells, the levels of pCx43 on the plasma membrane were significantly higher than control cells and cells treated with Tzb alone (62.97 ± 4.81; *p* < 0.005 versus control cells and *p* < 0.05 versus Tzb-treated cells).

## 3. Discussion

Breast cancer is the most common form of cancer in women, and it is estimated that there are around 1.4 million new cases every year [23,24]. A large segment (25–30%) of human breast cancers overexpress the protooncogene c-neu/c-erbB-2, which codes for the HER2 cell surface protein. HER2 receptor-positive breast neoplasms are more aggressive, are therefore more difficult to treat, and have a particularly poor prognosis [25]. Since the approval of the HER2 humanized monoclonal antibody Trastuzumab as a treatment, an important improvement in response rate and overall survival has been determined [26]. However, cardiac toxicity was recognized as an important side effect of Tzb, manifested as symptomatic CHF or asymptomatic LVEF decline, that in some cases can cause severe cardiac insufficiency and even death [27]. Because the number of patients who develop any type of cardiovascular disease increases significantly when Tzb is given in combination with Doxo [28], currently used clinical protocols involve their sequential administration. However, this protocol is still associated with a risk of cardiac dysfunction in up to one-quarter of breast cancer patients, along with an increased risk of arrhythmia [29]. 

The knowledge of the molecular mechanisms involved in the cardiotoxicity induced by the two chemotherapeutics could help to prevent and/or limit the progression of cardiac dysfunction. To this aim, we used an in vitro model to reproduce the effects of the sequential administration of Doxo and Tzb. The cell line used is H9c2 cells, a clonal cell line derived from rat heart that has shown many similarities to primary cardiomyocytes, including membrane morphology, G-protein signalling expression, electrophysiological properties, and constitutive expression of Cx43 [30] as well as expression of the HER2 receptor on the membrane [31]. This cellular model has been extensively used as an alternative in vitro model to cardiomyocytes to evaluate the cardiotoxic effects of chemotherapeutic agents and the molecular mechanisms involved [18,32,33].

As a first step, we analysed the effects of Doxo and Tzb sequential administration on the induction of oxidative stress and apoptosis in order to evaluate the effectiveness of our experimental model, designed on the basis of our previous studies showing the cardiotoxic effects of Doxo [34] and Tzb [17] alone.

Here, we confirmed that both Doxo and Tzb alone are able to induce cytosolic and mitochondrial oxidative stress, mitochondrial membrane depolarization, and thus apoptosis. Furthermore, our results showed that in our experimental conditions, Doxo administration is associated with a globally preserved intracellular Ca^2+^ content, while Tzb affects intracellular Ca^2+^ handling. Especially, a reduced mitochondrial Ca^2+^ uptake was observed in Doxo and Tzb sequentially treated cells. Beyond increased basal levels of Ca^2+^, in Doxo and Tzb sequentially treated cells, higher cytosolic and mitochondrial ROS production as well as mitochondrial membrane depolarization were observed. These results support the notion that when the Ca^2+^ levels increase in a prolonged and sustained manner, they prompt mitochondrial permeability transition pores, leading to dissipation of transmembrane potential and to increased permeability of the outer membrane. Consequently, the release of cytochrome C in the cytosol and the activation of the apoptotic pathway are reported [33]. In agreement with this, in our experimental model, increased levels of cytochrome c in the cytosol and of hypodiploid nuclei were detected.

These data are in line with previous studies showing that Tzb increases Doxo-induced cellular damages [35], although the molecular mechanisms underlying this effect are not well established [36,37]. Previous studies showed a rearrangement and altered expression of Cx43 and of pCx43 in cardiac cells in response to chemotherapeutic drugs, presumably as an adaptative response put in place by cardiac cells to counteract drug-induced cardiac damage [22,38]. Cx43 is a member of the Connexin family, integral membrane proteins that form gap junctions to enable the direct cytoplasmic exchange of information and small signalling molecules, such as Ca^2+^, between adjacent cells that contribute to cardiac conduction [39,40]. Furthermore, Cx43 is widely involved in cell survival and death signals transduction since it can spatially extend apoptosis through the communication of cell death signals from apoptotic cells to healthy cells [41]. Cx43 is a phosphoprotein having multiple phosphorylation sites. Phosphorylation of Cx43 by PKC on Ser368 induces a diminished hemichannel assembly, a decrease in gap junction intercellular communication, and a reduction in the half-life of the protein [42,43,44]. 

Thus, as a second step, in our study, we analysed the levels of Cx43 and pCx43 in Doxo and Tzb sequentially treated cells and we showed increased levels of pCx43 on the plasma membrane of cells sequentially treated with Doxo and Tzb, despite no evident alterations having been observed for Cx43. These results support the hypothesis of a defence strategy implemented by cells to counteract death spreading. It has been shown that when cells are exposed to hazardous stimulants, gap junction intracellular communication downregulation occurs, causing injured cells to lose rescue signals provided by healthy cells, which results in a loss of normal growth regulation by the surrounding cells and growth independence [45]. 

Furthermore, it has been reported that increased levels of intracellular calcium concentrations, as those observed in our experimental model, uncouple gap junctions in most tissues [46]. Intracellular calcium dysregulation is widely indicated as the onset of apoptosis [47]; thus, calcium gating of gap junctional intercellular channels might also constitute a protective mechanism in the heart, thus preventing the spreading of injury between neighbouring cells and attenuating leakage of metabolites [48].

Based on these observations, we can conclude that Doxo and Tzb sequential administration induces increased cytosolic and mitochondrial ROS production, calcium homeostasis dysregulation and thus apoptosis in cardiac cells. We hypothesize that the observed increased phosphorylation of Cx43 is an adaptative response to stress that tries to reduce the spread of death signals.

This hypothesis is in agreement with the theory termed as “healing over” by Engelmann in 1877. As he said, “cardiomyocytes live together but die alone” [49]. 

However, we can speculate that the drug-induced upregulation of pCx43 could be responsible for the conduction abnormalities associated with Doxo and Tzb co-administration.

In conclusion, this study reports that Doxo and Tzb sequential administration further increases oxidative stress and apoptosis, as well as Cx43 phosphorylation on Ser368, in cardiomyocytes, compared to the two drugs administered alone as previously reported by our research group [16,17,32,34,38]. These data further contribute to clarify the molecular mechanisms underlying the effects observed in clinical practice due to Doxo and Tzb sequential administration.

The main limitation of this work is that the observations were made in an in vitro system. Further in vivo studies are planned to better clarify if Cx43 could be a pharmacological target in Doxo and Tzb-induced cardiotoxicity.

## 4. Materials and Methods

### 4.1. Materials

Doxorubicin and Trastuzumab were purchased from Sigma (Milan, Italy). Embryonic rat heart cardiomyocyte-derived cell line H9c2 was purchased from the American Tissue Culture Collection (Manassas, VA, USA) and was grown to confluence in Dulbecco’s modified Eagle’s Medium (DMEM; Microgem, Naples, Italy) with 10% foetal bovine serum (FBS; Microgem) and antibiotics (25 U/mL penicillin and 25 U/mL streptomycin) under an atmosphere of 95% air/5% CO_2_ at 37 °C.

### 4.2. Experimental Protocol

H9c2 cells were treated according to this experimental model:

Control cells were kept in DMEM 10% FBS for 24 h.

Doxorubicin-treated cells were treated with Doxorubicin (1 µM) for 4 h in DMEM 10% FBS, and then the medium was replaced with a fresh one, and the cells were kept in DMEM 10% FBS for 20 h.

Trastuzumab-treated cells were kept in DMEM 10% FBS for 4 h, and then the medium was replaced, and Trastuzumab (200 nM) was added for 20 h.

Doxorubicin and Trastuzumab sequentially treated cells were treated with Doxorubicin (1 µM) for 4 h, and then the medium was replaced with a fresh one, and Trastuzumab (200 nM) was added for 20 h.

### 4.3. Measurement of Intracellular Reactive Oxygen Species (ROS)

ROS formation was evaluated using the probe 2′,7′-dichlorofluorescin diacetate (H_2_DCF-DA). H_2_DCF-DA is cleaved by intracellular esterases to form H_2_DCF that in the presence of intracellular ROS is rapidly oxidized to the highly fluorescent DCF. Briefly, H9c2 cells (4.5 × 10^5^ cells/well) were plated into 6-well plates and treated in agreement with the experimental protocol described above. Cells were then collected, washed twice with phosphate buffer saline (PBS) buffer and then incubated in PBS containing H_2_DCF-DA (10 μM) at 37 °C. After 45 min, cells fluorescence was evaluated using fluorescence-activated cell sorting (FACSscan; BD FacsCalibur, Milan, Italy) and analysed with CellQuest software.

### 4.4. Measurement of Mitochondrial Superoxide Formation

Mitochondrial superoxide formation was evaluated by means of MitoSOX Red, a fluorogenic dye that once targeted to mitochondria of living cells is readily oxidized by superoxide but not by other ROS-generating systems and exhibits red fluorescence [50,51]. For this analysis, H9c2 cells (4.5 × 10^5^ cells/well) were plated in 6-well plates and treated in agreement with the experimental protocol described above. After incubation period, MitoSOX Red (2.5 μM) was added for 15 min at 37 °C, and then cells were washed gently with PBS and collected for fluorescence evaluation by means of flow cytofluorometry. Cell fluorescence was evaluated using FACS scan and analysed by CellQuest software.

### 4.5. Measurement of Mitochondrial Membrane Depolarization

Mitochondrial membrane depolarization was evaluated by means of the fluorescent dye tetramethylrhodamine methyl ester (TMRE) that due to its positive charge, penetrates and accumulates in mitochondria in inverse proportion to membrane potential [52]. For these experiments, H9c2 cells (4.0 × 10^5^ cells/well) were seeded in 6-well tissue culture plates and treated in agreement with the experimental protocol described above. Cells were then collected, washed twice with phosphate buffer saline (PBS) buffer and then incubated in PBS containing TMRE (5 nM) at 37 °C. After 30 min, cells fluorescence was evaluated using a fluorescence-activated cell sorting and was analysed using CellQuest software.

### 4.6. Measurement of Intracellular Calcium Signalling

Intracellular calcium concentrations were measured using the fluorescent indicator dye Fura 2-AM, the membrane-permeant acetoxymethyl ester form of Fura 2. H9c2 cells (3 × 10^4^ cells/plate) were seeded in 86 mm tissue culture plates and treated as described above. After the incubation period, cells were washed in phosphate buffered saline (PBS) and resuspended in 1 mL of Hank’s balanced salt solution (HBSS) containing 5 μM Fura 2-AM for 45 min. Thereafter, cells were washed with the same buffer to remove excess Fura 2-AM and incubated in calcium-free HBSS/0.5 mM EGTA buffer for 15 min to allow hydrolysis of Fura 2-AM into its active-dye form, Fura 2. H9c2 cells then were transferred to the spectrofluorimeter (Perkin-Elmer LS-55; Waltham, MA, USA). Treatment with Ionomycin (1 μM final concentration), or with carbonyl cyanide p-trifluoromethoxy-pyhenylhydrazone (FCCP, 50 nM final concentration) was carried out by adding the appropriate concentrations of each substance into the cuvette in calcium-free HBSS/0.5 mM EGTA buffer. The excitation wavelength was alternated between 340 and 380 nm, and emission fluorescence was recorded at 515 nm. The ratio of fluorescence intensity of 340/380 nm (F340/F380) is strictly related to intracellular free calcium, as previously reported [51,53]. Data were expressed as percentage of delta (% Δ) increase in fluorescence ratio (F340/F380 nm) induced by Ionomycin (1 μmol/L) or FCCP (0.05 μmol/L)—basal fluorescence ratio (F340/F380 nm)/basal fluorescence ratio (F340/F380 nm).

### 4.7. Analysis of Apoptosis 

H9c2 cells (4.5 × 10^5^ cells/well) were plated in a 6-well plate and treated in agreement with the experimental protocol described above. Cells were then washed twice with PBS and incubated in 500 μL of a solution containing 0.1% Triton X-100, 0.1% sodium citrate and 50 μg/mL Propidium Iodide (PI), at 4 °C for 30 min in the dark. The PI-stained cells were analysed by means of FACS using CellQuest software (version 5.2.1). Data are expressed as the percentage of cells in the hypodiploid region.

### 4.8. Flow Cytometry Analysis

H9c2 cells were cultured at a density of 4.5 × 10^5^ cells/well in a 6-well plate and were allowed to grow for 24 h and treated as described in order to assess cytochrome c, Connexin 43 (Cx43) and Connexin 43 phosphorylated on Ser368 (pCx43) expression. For cytochrome c detection, cells were collected with scraper and treated with fixing buffer (containing 4% formaldehyde, 0.1% NaN_3_ and 2% FBS in PBS) for 20 min and then permeabilized with fix perm buffer (fixing buffer containing 0.1% Triton X) for 30 min, and then anti-cytochrome *c* antibody was added. Anti-rabbit FITC antibody was used as a secondary antibody (eBioscience, San Diego, CA, USA) [17,52]. For Cx43 and pCx43 detection, cells were treated with fixing buffer for 20 min; subsequently, cells were incubated with Cx43 and pCx43 antibody (all antibodies were from Santa Cruz Biotechnologies), and anti-rabbit or anti-mouse FITC antibody was used as a secondary antibody (eBioscience, San Diego, CA, USA), for 1 h at 4 °C. Cells collected were evaluated by fluorescence-activated cell sorting (FACSscan; Becton Dickinson, Franklin lakes, NJ, USA) and data obtained were analysed by means of CellQuest software. Results are shown as percentage of positive cells.

### 4.9. Statistical Analysis

Statistical analysis was performed with the aid of commercially available software GraphPad Prism7 (GraphPad Software Inc., San Diego, CA, USA). Results are presented as the mean ± S.E.M. for at least three independent experiments, each performed in duplicate. Statistical analysis was performed by an analysis of variance test, and multiple comparisons were made by Bonferroni’s test. A *p* value lower than 0.05 was considered significant.

## Figures and Tables

**Figure 1 ijms-23-06375-f001:**
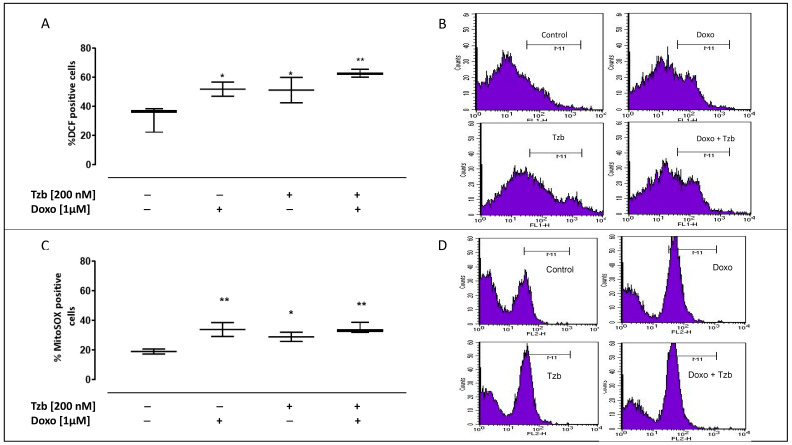
Effect of Doxo and Tzb and their association with cytosolic and mitochondrial ROS production. Doxo (1 µM) was administered for 4 h, and Tzb (200 nM) was administered for 20 h. ROS formation was evaluated by means of 2′,7′dichlorofluorescein diacetate (H2DCF-DA) probe in H9c2 cells. ROS production is expressed as mean ± SEM of percentage of DCF positive cells of at least three independent experiments, each performed in duplicate (**A**). Mitochondrial superoxide production was measured by means of the probe MitoSOX Red in H9c2 cells through flow cytometry analysis, and it is expressed as mean ± S.E.M. of MitoSOX positive cells percentage of at least three independent experiments each performed in duplicate (**C**). Data were analysed by variance test analysis, and multiple comparisons were made by Bonferroni’s test. * *p* < 0.05 and ** *p* < 0.005 vs. non-treated cells. Panels (**B**,**D**) report representative histograms for the flow cytometry analysis of DCF and MitoSOX, respectively.

**Figure 2 ijms-23-06375-f002:**
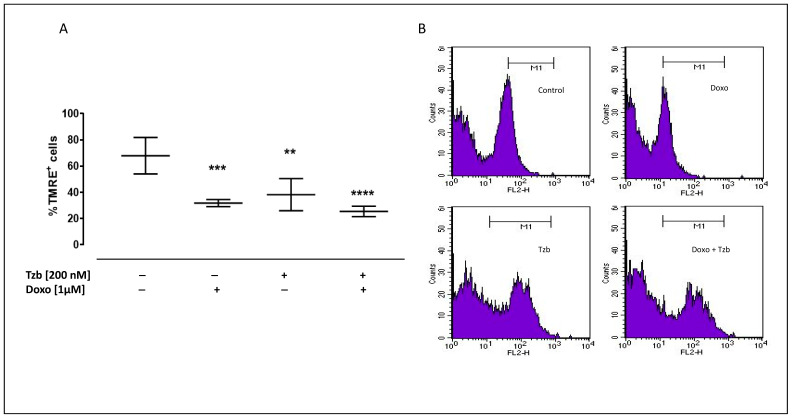
Doxo and Tzb and their association effect on mitochondrial membrane depolarization. Doxo (1 µM) was administered for 4 h, and Tzb (200 nM) was administered for 20 h. Mitochondrial membrane potential was evaluated by flow cytometry analysis with Tetramethylrhodamine ethyl ester (TMRE), a cell permeant, positively charged, red-orange dye, which comes in and accumulates in the mitochondria in inverse proportion to the membrane potential. The low value of TMRE+ cells percentage means indicates that the TMRE dye was not trapped in the mitochondrial membrane due to its depolarization. Results are expressed as mean ± S.E.M. of fluorescence intensity of at least three independent experiments, each performed in duplicate (**A**). Data were analysed by variance test analysis, and multiple comparisons were made by Bonferroni’s test. ** *p* < 0.005, *** *p* < 0.001 and **** *p* < 0.0001 vs. non-treated cells. Panels (**B**) reports representative histograms for the flow cytometry analysis.

**Figure 3 ijms-23-06375-f003:**
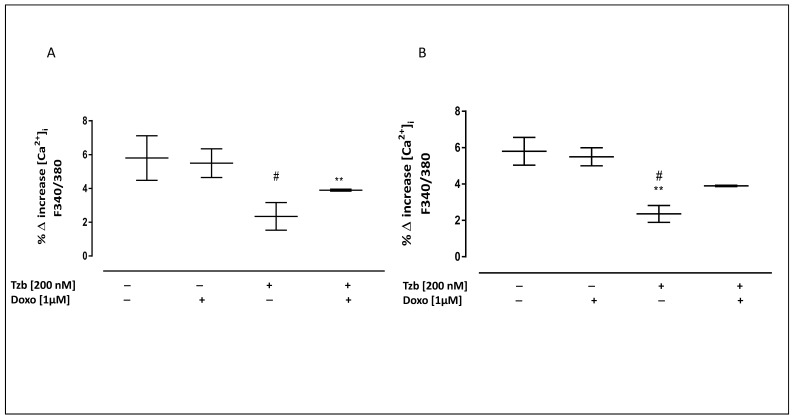
Effect of Doxo and Tzb and their association with calcium homeostasis. Doxo (1 µM) was administered for 4 h, and Tzb (200 nM) was administered for 20 h. Mitochondrial calcium pool was evaluated on H9c2 cells in calcium-free medium in the presence of FCCP (50 nM) (panel (**A**)). Panel (**B**) shows the intracellular calcium content in calcium-free medium evaluated by means of Ionomycin (1 μM). Results are expressed as mean ± S.E.M. of percentage of delta (% Δ) increase in FURA 2 ratio fluorescence (340/380 nm) from at least three independent experiments, each performed in duplicate. Data were analysed by variance test analysis, and multiple comparisons were made by Bonferroni’s test. ** *p* < 0.005 vs. non-treated cells, # *p* < 0.05 vs. Doxo-treated cells.

**Figure 4 ijms-23-06375-f004:**
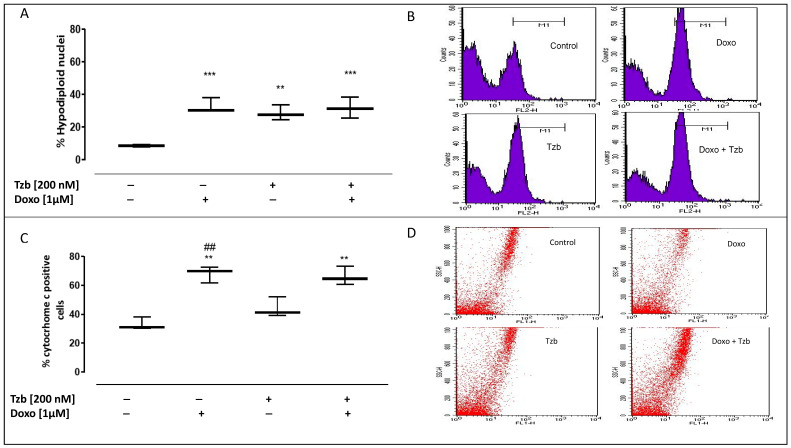
Doxo and Tzb and their association effect on apoptosis. Doxo (1 µM) was administered for 4 h, and Tzb (200 nM) was administered for 20 h. H9c2 cells were stained by PI, and individual nuclei fluorescence was measured by flow cytometry. Results are expressed as mean ± S.E.M. of percentage of hyplodiploid nuclei from at least three independent experiments, each performed in duplicate (panel (**A**)). Panels (**B**) reports the representative histograms for the flow cytometry analysis. Cytosolic cytochrome c level was detected by flow cytometry analysis. Results are expressed as mean ± S.E.M. of percentage of cytochrome c positive cells from at least three independent experiments each performed in duplicate (panel (**C**)). Data were analysed by variance test analyses, and multiple comparisons were made by Bonferroni’s test. ** *p* < 0.005 and *** *p* < 0.001 vs. non-treated cells, ## *p* < 0.005 vs. Doxo-treated cells. Panel (**D**) reports the representative images of flow cytometry analysis of cytochrome c expression.

**Figure 5 ijms-23-06375-f005:**
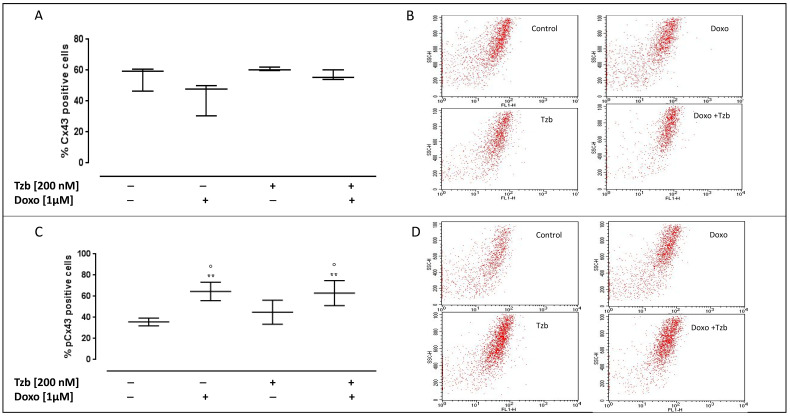
Doxo and Tzb and their association effect on Cx43 and pCx43 expression. Doxo (1 µM) was administered for 4 h, and Tzb (200 nM) was administered for 20 h. Cx43 and pCx43 expression was checked through flow cytometry analysis. Results are expressed as mean ± S.E.M. of percentage of Cx43 (**A**) and pCx43 (**C**) positive cells from at least three independent experiments, each performed in duplicate. Data were analysed by variance test analysis, and multiple comparisons were made by Bonferroni’s test. ** *p* < 0.005 vs. non-treated cells, ° *p* < 0.005 vs. Tzb-treated cells. Panels (**B**,**D**) report representative images of flow cytometry analysis of Cx43 and pCx43 expression, respectively.

## Data Availability

The data presented in this study are available on request from the corresponding author.

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
