# Peer review of "Trastuzumab and Doxorubicin Sequential Administration Increases Oxidative Stress and Phosphorylation of Connexin 43 on Ser368"

_ijms, 2022, doi:10.3390/ijms23126375_

Round 1
Reviewer 1 Report
The authors are presenting information regarding the impact of 2 chemotherapies on cardiomyocytes. They have done 3 indep experiments and there seem to be significant findings. Given the impact of these medications on patients it will be helpful to better explain what they do to these cells. Some aspects of this paper have been previously reported by this group such as some of the cx43 data, but they indicate how this paper is unique.
Minor typos:
- separate words on line 14 cardiomyoblasts were not cardiomyoblastswere
- define CHF line 46
- line 212 previous not prvious
Author Response
Answers to Referee 1
The authors are presenting information regarding the impact of 2 chemotherapies on cardiomyocytes. They have done 3 indep experiments and there seem to be significant findings. Given the impact of these medications on patients it will be helpful to better explain what they do to these cells. Some aspects of this paper have been previously reported by this group such as some of the cx43 data, but they indicate how this paper is unique.
Reply: Thank you for your appreciation.
In this revised version we inserted in the Discussion section information concerning the cellular model we used in this study. In particular we used these cells because they are extensively used as a model to study cardiomyocites response. In particular, they have been also used, both by our research group and others, to evaluate the cardiotoxic effects of chemotherapeutic agents and the molecular mechanisms involved [18, 32-33].
- Horie, T. et al., Cardiovasc Res. 2010. 87 (4): 656-64.
- Pecoraro, M. et al., Cardiovasc Toxicol. 2015. 15 (4): 366-76.
- Ma, W. et al., Front Cell Dev Biol. 2020. 8: 434.
For other references, please, see:
Dallons et al., Front Pharmacol. 2020. 11: 79
Wang et al., Oncology Letters. 2018. 15 (1): 839-49
Boutbir et al., Eur Cardiol. 2020. 15: e33
Zhang et al., Drug Design, Development and Therapy. 2021. 15: 87-97
Furthermore, to better clarify the novelty of this study in the Discussion section we inserted the sentence “In conclusion, this study reports that Doxo and Tzb sequential administration further increase oxidative stress and apoptosis, as well as Cx43 phosphorylation on Ser368, in cardiomyocytes compared to the two drugs administered alone as previously reported by our research group [16,17,32,34,38]. These data further contribute to clarify the molecular mechanisms underlying the effects observed in clinical practice cue to Doxo and Tzb sequential administration.”16. Pecoraro, M. et al., Int J Mol Med. 2020, 46 (3): 1197-1209.
- Pecoraro, M. et al., Toxicol In Vitro 2020, 67: 104926.
- Pecoraro, M. et al., Cardiovasc Toxicol. 2015. 15 (4): 366-76.
- Pecoraro, M. et al., Int J Mol Sci. 2017. 18 (10): 2121.
- Pecoraro, M. et al., Int J Mol Sci 2018, 19: 757.
Minor typos:
- separate words on line 14 cardiomyoblasts were not cardiomyoblastswere
Reply; the words have been separated
- define CHF line 46
Reply: CHF has been now defined correctly (chronic heart failure)
- line 212 previous not prvious
Reply: the wrong word has been now corrected.
The paper has been enterily revised and typos have been corrected.

Reviewer 2 Report
Authors need to provide pictures such as immunostaining, there are only analysis data. Authors did not provide adequate information for reviewers to make decision.
Author Response
Authors need to provide pictures such as immunostaining, there are only analysis data.
Reply: The figures 1, 2, 4 and 5 are results of analysis obtained by FACS evaluation. Thus, as suggested, we inserted representative images of FACS analysis for each evaluated mediators. We hope that these added pictures contribute to clarify the results.
Authors did not provide adequate information for reviewers to make decision.
Reply: We are sorry that the previous version didn’t provide sufficient information. In order to give additional information concernig our research, we extensively revised the Discussion section; in particular, we gave a particular importance to the hypothesis and to the conclusion of the study.

Round 2
Reviewer 2 Report
1. Please provide immunostaining images for Fig 4 and 5, cytometry only tell us the expression intensity but not the location
2. Please provide the images of calcium hemostatsis picture using FCCP on Figure 3
For example:
- DOI:
- 10.1371/journal.pone.0080574 Figure 1
Author Response
- Please provide immunostaining images for Fig 4 and 5, cytometry only tell us the expression intensity but not the location.
Reply: Regarding your request this would require new experiments using immunostaining tecniques, unfortunately, due to technical problems we cannot perform them at the moment, and we do not have the possibility to establish a collaboration with other colleagues in a short time, considering the deadline for the submission of the revised version of the manuscript.
However, in this study we used Cytofluorimetry evaluation for data reported in figure 4 since this method has been previously used by our group to evaluate cytosolic levels of Cytochrome C (Pecoraro et al., Toxicology in Vitro, 2018, 47: 120-128; Pecoraro et al., Toxicology in Vitro, 2020, 67: 104926).
Indeed, Campos and co-workers in an elegant paper reported that cytofluorimetry evaluation for cytocrome c show more advantages compared to Western blot and fluorescence microscopy since it increases both accuracy and the number of cells analyzed during apoptosis (Campos et al., Cytometry 2006 Jun;69(6):515-23.).
Concerning the Cx43 and pCx43 evaluation by cytofluorimetry (data reported in figure 5) this method has been previoulsly used to detect membrane proteins.
Detection of membrane proteins Cx43 and pCx43 (data reported in figure 5) has been performed in agreement with published data.
- Hogg et al., Methods 2015; 82: 38-46
- Torres-Martínez et al., Exp Cell Res. 2017; 350 (1): 226-235.
- Gonzalez-Villalva et al., Toxicol Ind Health. 2016 May; 32 (5): 908-18.
- Kobusiak-Prokopowicz Met al., Kardiol Pol. 2006 Oct; 64 (10): 1094-100; discussion 1101.
- Ye et al., Ann Clin Lab Sci. 2017 Aug; 47 (4): 389-394.
- Graf et al., Eur J Haematol. 2005 Dec; 75 (6): 477-84.
In this occasion we are sorry with reviewer because there was a mistake in Materials and methods section. From the re-reading of the text we have seen that in the Materials and methods section it was wrongly indicated that for the detection of Cx43 and pCx43 the cells had been permeabilized by means of a buffer containing 4% formaldehyde, 0.1% NaN3, 2% FBS and 0.1% Triton X). This method has been used only for the detection of cytosolic Cytochrome C. Indeed, to analyse the levels of Cx43 and pCx43 on plasma membrane the cells were not permeabilized to prevents antibodies from entering the cells. Thus, our results on Cx43 and on pCx43 only refer to membrane expression.
We apologize if this error, due to an incorrect revision, has generated doubts about the localization of the analyzed proteins.
The correct procedure used has now been reported in the Materials and method section as follow: “H9c2 cells were cultured at a density of 4.5 x 105 cells/well in a 6-well plate and allow to grow for 24 hours and treated as described in order to assess Cytochrome c, Connexin 43 (Cx43) and Connexin 43 phosphorylated on Ser368 (pCx43) expression. For Cytochrome c detection, cells were collected with scraper and treated with fixing buffer (containing 4% formaldehyde, 0.1% NaN3 and 2% FBS in PBS) for 20 minutes and then permeabilized with fix perm buffer (fixing buffer containing 0.1% Triton X) for 30 minutes and then anti-cytochrome c antibody was added. Anti-rabbit FITC antibody was used as a secondary antibody (eBioscience, CA, USA) [17, 52]. For Cx43 and pCx43 detection cells were treated with fixing buffer for 20 minutes, subsequently, cells were incubated with Cx43 and pCx43 antibody (all antibodies were from Santa Cruz Biotechnologies) and anti-rabbit or anti-mouse FITC antibody was used as a secondary antibody (eBioscience, CA, USA), for 1 hour at 4°C. Cells collected were evaluated by fluorescence-activated cell sorting (FACSscan; Becton- Dickinson) and data obtained were analyzed by means of Cell Quest software. Results were shown as percentage of positive cells.”
We hope that the corrections have been useful in clarifying the procedures used.
- Please provide the images of calcium hemostatsis picture using FCCP on Figure 3
For example: DOI: 10.1371/journal.pone.0080574 Figure 1
Reply: We thanks the reviewer for this suggestion and we don’t exclude to use this method un the future. However, for our study we preferred to use the same methods used for our previous paper concerning the same topic of research (TUOI). This method has been extensively used also by other authors (ALTRI) Unlike the suggested work, in our study, the measurement of intracellular Ca2+ signaling was performed by spectrofluorimetric analysis using the probe Fura 2-AM (Sigma).
More specifically, after treatments, cells were loaded with the probe suspended in Ca2+-free Hanks balanced salt solution at a final concentration of 5 µM at 37 °C for 45 min with a subsequent 15 min washout. Then, cells were transferred in quarz cuvette and analysed at the spectrofluorimeter. During the analysis the cells were kept stirred with a magnet. Basal Ca2+ level (F340/F380 nm ratio) was acquired after 200 s of stabilization, then FCCP (50 nM final concentration) was added and the F340/F380 nm ratio was acquired. Ionomicyn (1 µM final concentration) was added 200s after FCCP administration and the F340/F380 nm ratio was acquired.
For each sample, mitochondrial calcium content was evaluated as follow: [(F340/F380 nm after FCCP administration - F340/F380 nm basal) / F340/F380 nm basal] x 100 and expressed as % Δ increase of Fura 2 ratio fluorescence (F340/F380 nm). Cytosolic calcium content was evaluated as: [(F340/F380 nm after Ionomicyn administration - F340/F380 nm basal) / F340/F380 nm basal] x 100 and expressed as % Δ increase of Fura 2 ratio fluorescence (F340/F380 nm). For this reason, as also done in our previous publications, only quantitative and not qualitative data are reported.
This method has been extensively used by us and other authors:
- Tiribuzi R. et al., Arch Oral Biol. 2014 Dec; 59(12):1377-83.
- Villalba M. et al., J Biol Chem. 1994 Jan 28;269(4):2468-76.
- Improta-Brearset al., Proc Natl Acad Sci U S A 1999 Apr 13; 96 (8): 4686-91.
- Pecoraro M. et al., Int J Mol Sci. 2021 Oct 27;22(21):11599.
- Pecoraro M. et al., Int J Mol Sci. 2018 Mar 7;19(3):757.
- Pecoraro M. et al., Int J Mol Sci. 2017 Oct 11;18(10):2121.
- Pecoraro M et al., Toxicol In Vitro. 2020 Sep; 67:104926.
- Pecoraro M. et al., Cardiovasc Toxicol. 2015 Oct;15(4):366-76.
- Pecoraro M. et al., Toxicol In Vitro. 2018 Mar; 47: 120-128.
- Sorrentino R. et al., Am J Pathol. 2015 Nov;185(11):3115-24.
- Popolo A. et al., Can J Physiol Pharmacol. 2011 Jan;89(1):50-7.
- Bizzarro V. et al., PLoS One. 2012;7(10): e48246.
- González A. et al., Brain Res. 2007 Oct 31; 1178: 28-37.

Round 3
Reviewer 2 Report
Please include immunostaining pictures and calcium studies
Author Response
We understand that immunofluorescence images could help make our work more elegant and could corroborate our data. However, as we previously said, we cannot add immunostaining pictures at now, for technical problems. Furthermore, we want to emphasize that the methods used in this work have been established on the basis of our previous studies, and also other authors have published many papers aplying them.
For Cytochrome c, please see:
- Campos et al., Cytometry 2006 Jun;69(6):515-23.
- Pecoraro et al., Toxicology in Vitro, 2018, 47: 120-128;
- Pecoraro et al., Toxicology in Vitro, 2020, 67: 104926
For membrane proteins Cx43 and pCx43, please see:
- Hogg et al., Methods 2015; 82: 38-46 (they used cytofluorimetry to detect the membrane transporters OATP1B1 and P-gp);
- Torres-Martínez et al., Exp Cell Res. 2017; 350 (1): 226-235 (they used cytofluorimetry to detect the membrane protein Claudin-1)
- Kobusiak-Prokopowicz Met al., Kardiol Pol. 2006 Oct; 64 (10): 1094-100; discussion 1101 (they used cytofluorimetry to detect the membrane protein P-selectin);
- Ye et al., Ann Clin Lab Sci. 2017 Aug; 47 (4): 389-394 (they used cytofluorimetry to detect the membrane transporters OATP1B1 and P-gp.
- Graf et al., Eur J Haematol. 2005 Dec; 75 (6): 477-84 (they used cytofluorimetry to detect the membrane receptors PRR1 and PRR2).
For Calcium levels analysis, please, see our previous reports:
- Pecoraro M. et al., Int J Mol Sci. 2021 Oct 27;22(21):11599.
- Pecoraro M. et al., Int J Mol Sci. 2018 Mar 7;19(3):757.
- Pecoraro M. et al., Int J Mol Sci. 2017 Oct 11;18(10):2121.
- Pecoraro M et al., Toxicol In Vitro. 2020 Sep; 67:104926.
- Pecoraro M. et al., Cardiovasc Toxicol. 2015 Oct;15(4):366-76.
- Pecoraro M. et al., Toxicol In Vitro. 2018 Mar; 47: 120-128.
- Sorrentino R. et al., Am J Pathol. 2015 Nov;185(11):3115-24.
- Popolo A. et al., Can J Physiol Pharmacol. 2011 Jan;89(1):50-7.
And furthermore:
- Tiribuzi R. et al., Arch Oral Biol. 2014 Dec; 59(12):1377-83.
- Villalba M. et al., J Biol Chem. 1994 Jan 28;269(4):2468-76.
- Improta-Brearset al., Proc Natl Acad Sci U S A 1999 Apr 13; 96 (8): 4686-91.
- Bizzarro V. et al., PLoS One. 2012;7(10): e48246.
- González A. et al., Brain Res. 2007 Oct 31; 1178: 28-37.
We really appreciated that you now indicate that the research design is appropriate, the methods are adequately described and, moreover, that the conclusions are supported by the results. So, we hope that, despite you said that the results could be clearly presented, this doesn't make the paper any less interesting for readers.
